# GNN-BASED REINFORCEMENT LEARNING AGENT FOR SESSION-BASED RECOMMENDATION

## ABSTRACT

This paper focuses on session-based item recommendation and the challenges of using Reinforcement Learning (RL) in recommender systems. While traditional RL methods rely on one-hot encoded vectors as user state, they often fail to capture user-specific characteristics, which may provide misleading results. In contrast, recently, Graph Neural Networks (GNNs) have emerged as a promising technique for learning user-item representations effectively. However, GNNs prioritize static rating prediction, which does not fully capture the dynamic nature of session-based recommendations. To address these limitations, we propose a novel approach called GNN-RL-based Recommender System (GRRS), which combines both frameworks to provide a unique solution for the session-based recommendation [1]. We demonstrate that our method can leverage the strengths of both GNNs and RL while overcoming their respective shortcomings. Our experiments on several logged public datasets validate the efficacy of our approach over various SOTA algorithms. Additionally, we offer a solution to the *offline training problem*, which is often encountered by RL algorithms when employed on logged datasets, which may be of independent interest.

## 1 INTRODUCTION

Recommender systems are vital in e-commerce, online movie streaming, and many other web platforms. With the explosive increase in the number of users and items, designing an efficient and effective recommender agent is crucial. Traditional methods like matrix factorization aimed to exploit the hidden structures, like low-rank, in the partially observed user-item reward matrix (see (Gopalan et al., 2016; Sen et al., 2016) and references therein) and follow a greedy strategy to recommend items. These existing methods are suboptimal as they fail to capture the dynamic nature of evolving user preferences over time. Therefore, it is crucial to adopt session-based recommender systems to effectively address these challenges.

In literature, the session-based recommendation was first introduced in (Hidasi et al., 2015), which deals with recommending a set of items to a user that he/she is most likely to consume in a *session*. One can define a *session* as a set of user-item interactions during which the user's preference is assumed to be stationary. This setting is highly relevant in e-commerce and over-the-top (OTT) media platforms and imposes distinct challenges as user behaviour may change across sessions. To address this, some works have employed Recurrent Neural Networks Hidasi et al. (2016) and its variants such as Hidasi et al. (2015); Dabral et al. (2023). Furthermore, collaborative filtering techniques Su & Khoshgoftaar (2009) have been used to model the evolving user tastes by keeping track of the history of individual users and performing clustering on the 1-0 encoded vectors to group similar users together. This can help in generalizing recommendations across users with similar buying/watched/liked histories. A potential stumbling block for these methods arises when there are users with similar tastes (e.g., users who like products of a similar genre) but with no common liked items (i.e., no *long-range user-association*). This can lead to unpleasant recommendations followed by misleading user preference modeling, adversely affecting the user's long-term engagement with the system. We discuss this problem in detail with a motivating example in Sec.3.

Recently, GNN He et al. (2020) based embeddings have proven effective in capturing long-term user association. In addition, GNN-based features are compact and can scale easily compared to

---

[1]Code available at `https://anonymous.4open.science/r/iclr24_gnn_rl/`

1-0 encoded feature vectors, whose size grows with the number of items. Thus, GNN is a unique method to generalize and combine collaborative and content-based filtering methods. However, in general, GNNs do not capture the *inherent* sequential nature of session-based recommendation systems Zhang et al. (2021). In order to address this challenge, preliminary approaches such as Zhang et al. (2021); Chang et al. (2023) have been proposed to make the GNNs more apt for sequential recommendation, but these methods are computationally expensive.

On the other hand, Reinforcement Learning (RL) is well-known to handle sequential data. In particular, when we model the user-item interaction system as a discounted infinite-state Markov Decision Process (MDP), which efficiently takes care of long-term user behaviour. Historically, multiarmed bandits (single-state RL) Li et al. (2010); Shi et al. (2023) followed by full-state RL-based algorithms like Deep-Q Learning, Actor-critic, Proximal Policy Optimization Lin et al. (2023), and similar other methods have been applied for the task of item-recommendation and have performed reasonably successfully. However, state representation always plays a vital role in all these methods.

In this paper, we propose a novel approach to leverage features of RL and GNNs to effectively extract the state representations and improve the item recommendations. Furthermore, state representations' importance is given in Sec. 3 to illustrate how the better state representations lead to better recommendations. Additionally, we devise a strategy to handle the *offline training problem* from which many RL-based recommender systems often suffer, leading to poor recommendations.

## 1.1 CONTRIBUTIONS

Our contribution can be summarized as follows:

- We propose a Graph Neural Network-based Reinforcement Learning Agent for the task of session-based item-recommendation, which we call GRRS (Alg. 1 and Alg.3). This combination of the previously well-studied frameworks *for* the recommendation task has been successfully implemented for the first time to the best of our knowledge.
- We provide a novel method to tackle the infamous "offline training problem" Prudencio et al. (2023) faced by standard RL-based recommendation systems, which can be of independent interest. Towards this, we utilize the *off-policy* nature of the DQN algorithm Sutton et al. (2000), along with the properties of the underlying Markov Decision Process (MDP) to create an *artificial experience replay buffer* (ARB). This helps to eradicate the troublesome *distributional shift* accompanying many RL algorithms working on static datasets (Sec.4).
- To reinforce our claims, we apply proposed methods to various standard datasets, viz., MovieLens-1m, MovieLens-100K Harper & Konstan (2015) and Goodreads Wan & McAuley (2018) and show its superiority over SOTA algorithms. In addition to the version presented in Alg1 and Alg.3, we perform various modifications thereof to improve the time complexity and compare all the results with each other (Sec.5). We also perform ablation studies to show the need and efficacy of our approach, which is essentially a *two-step* process.

## 1.2 ORGANIZATION

The paper is organized as follows: Sec.2 introduces the problem setting and the MDP model, Sec. 3 provides the motivation behind this work and the reason we choose to combine two well-known recommendation techniques, Sec.4 shows our algorithm (1) with network warming (Alg. 1), and (2) the fully adaptive scheme (Alg. 3). Sec.5 presents the simulations we carry out on logged datasets, and we conclude in Sec. 6.

## 2 PROBLEM SETUP

We consider the problem of session-based recommendation, where there are $M$ users to which there are $N$ items to be displayed. In session-based recommendations, user-item interactions are arranged temporally in the form of sessions/episodes. The learning agent can observe the history of the arriving user (and of previous users) $\mathcal{H}_t$ (which constitutes the past user ids and their watched histories) and use this information to predict a set of items which the user is most likely to consume in the next-session. Recommending favourable items to users maximizes user satisfaction, leading to increased long-term revenue extraction for the learning agent.

Towards this, we model the problem as a discounted-MDP ($\mathcal{M}(\mathcal{S}, \mathcal{A}, \mathbb{P}, r, \gamma)$). The state space $\mathcal{S}$ is defined as a subset of $\mathbb{R}^d$, where every vector in $\mathcal{S}$ represents the state of the system. The action space $\mathcal{A}$ is the set of all items that are available to be displayed to the arriving user. $\mathbb{P}(.|s,a)$ is defined as the transition probability kernel, which gives a probability distribution over the next state given the current state $s \in \mathcal{S}$ and action $a \in \mathcal{A}$. $r(s, a, s')$ denotes the immediate reward that the agent receives upon playing action $a$, in state $s$ and transitioning to state $s'$. $\gamma$ denotes a discount factor and is a number in $[0, 1)$. The discount factor is used to weigh the future rewards.

The dynamics of the transition kernel are assumed to be deterministic, as will be evident in Sec.4 where we present our method in full detail. In particular, we assume that the next state $s'$ of the user upon an action $a$ can be simulated via a simple weighted averaging scheme (eq.2). We find that doing this helps introduce a temporal evolution of the user state, which helps capturing the non-stationary nature of user preferences over time.

## 3 Motivation and Background

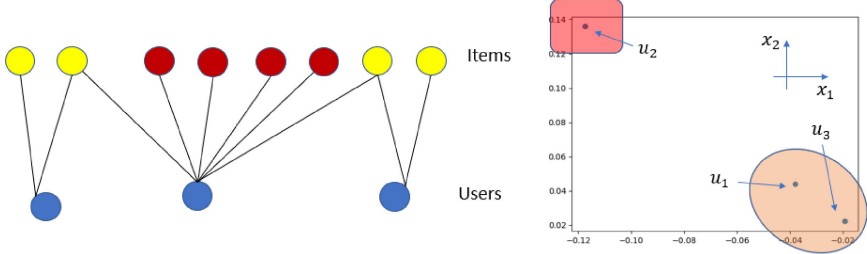

Figure 1: A toy example to illustrate the advantage of using GNN-features as states.

In this section, we describe a scenario where one-hot features based on user histories fail to capture the underlying structure in user preferences, leading to poor recommendations. We consider a toy example with 3 users and 9 items (see Fig. 1). The figure shows all the interactions until the current time step $t$. The nodes marked yellow represent items of genre type 1. The nodes marked red represent items of genre type 2. Consider the features representing user 1 and user 3. Clearly, the preferences of users 1 and 3 match closely as they are attracted towards items of the same genres. However, if one-hot encoded feature vectors (based on items watched) are used, then at time step $t + 1$, $\langle s_{u_1}^{one-hot}, s_{u_3}^{one-hot} \rangle = 0$, but $\langle s_{u_1}^{one-hot}, s_{u_2}^{one-hot} \rangle > 0$. Hence, this may lead to recommending unfavourable items to user 1 (and similarly to user 3). On the other hand, GNN embeddings (features) are calculated with the following equation.

$$f_A^{k+1} := \sum_{B \in Nbhd(A)} \frac{1}{\sqrt{|Nbhd(A)|}\sqrt{|Nbhd(B)|}} f_B^k, \quad k = 1, 2, \ldots, K \tag{1}$$

where $A$ and $B$ denote arbitrary nodes in the user-item graph. $Nbhd(A)$ ($Nbhd(B)$) denotes the nodes connected to the node $A$ ($B$). Clearly, if GNN-based embeddings are used as state vectors, one obtains a positive correlation between $s_1^{GNN}$ and $s_3^{GNN}$ with the value of number of hops $K \geqslant 4$, which would aid in reasonable item recommendation.

In order to reinforce this point, we show in Fig. 1 the node embeddings of the three users after passing through the GNN block. We see that users 1 and 3 appear proximally closer to each other as compared to user 2. This leads to better state approximation than their one-hot counterpart, which in turn would be helpful for recommendation when passed through the RL block next.

To formally see the effect of proximity of states on the quality of the recommendation, we state a result that holds under the following assumptions.

**Assumption 1.** *There exists a $L_r > 0$, such that for all states $s, s' \in \mathcal{S}$, $a \in \mathcal{A}$, we have $|r(s, a) - r(s', a)| \leqslant L_r \|s - s'\|_2$.*

**Assumption 2.** *There exists a $L_P > 0$, such that for all $s, s', \tilde{s} \in \mathcal{S}$, $a \in \mathcal{A}$, we have $|\mathbb{P}(s'|s, a) - \mathbb{P}(s'|\tilde{s}, a)| \leqslant L_P \|s - \tilde{s}\|_2$.*

These assumptions define a *smooth* MDP, where a change in the states does not lead to drastic changes in the reward functions and the transition kernels. With these assumptions, we make the following assertion.

**Theorem 1.** *With assumptions 1 and 2, for any $s, \tilde{s} \in \mathcal{S}$ and $a \in \mathcal{A}$, the following continuity property for the optimal state-action value $Q^*$ holds true:*

$$|Q^*(s, a) - Q^*(\tilde{s}, a)| \leqslant L_Q \|s - \tilde{s}\|_2,$$

*where, $L_Q$ is a problem-specific constant.*

*Proof Sketch.* The proof is a simple consequence of assumptions 1 and 2, followed by some elementary properties of discounted MDPs. The details are deferred to the appendix. □

**Note 1.** *Assume that the actual (unknown) state of the system is $s$ and that given by the pre-trained-GNN block be $\tilde{s}^{GNN}$ and by the one-hot vector encoding be $\tilde{s}^{one-hot}$ (generated by the movies watched by the current user). Let us also assume that all of these vectors belong to $\mathbb{R}^d$ for suitably large $d$. By the above result, the closer the actual state is to its approximation, the closer the corresponding optimal $Q-$values for the real and the approximated states will be. Hence, it is reasonable to expect Q-value estimates with better state approximation would lead to better recommendations.*

## 4 ALGORITHM

This section presents the GNN-RL-based Recommender System (GRRS) scheme in Alg. 1. We explain its individual blocks in detail below.

### 4.1 GNN BLOCK

We use *Graph Convolutional Network* (GCN) as a feature extractor as described in Sec3. In particular, we use the LightGCN version of the graph neural network, introduced in He et al. (2020), a computationally lighter version of the GCN. However, LightGCN performs empirically similarly to the full-blown version with much less computational and training complexity. Most of the GNN-based recommender systems employed in literature are employed for batch operations where the main tasks considered are edge prediction and matrix completion van den Berg et al. (2017); Zhang et al. (2022); Zhang & Chen (2019); Shen et al. (2021).

Depending on the computational resources available, we show two schemes that use the GNN and RL blocks in different ways.

- **With network warming.** We depict this scheme in Fig. 3. Here, we pre-train the GNN block over the train data offline. This gives a feature vector of every node (users and items) in the training graph (i.e., the user-item interaction graph for the train data). We call this the embedding matrix E. We do this pretraining in *batch-mode*. The loss function to be minimized in this pretraining is chosen as the Bayesian Personalized Ranking (BPR) loss He et al. (2020), which encourages observed user-item predictions to have increasingly higher values than unobserved ones.

  After obtaining E, a straightforward way to use it is to assign each node in the graph its embedding vector. However, by doing this, we lose the sequential nature of the problem and do not capture the dynamics of the user state evolution as she/he is shown items sequentially. In order to describe the evolving nature of the user state, we employ a simple technique shown in Alg.2, which is essentially a weighting and summing of the different item embeddings that the user has consumed in the past. One can also choose to sum any number $0 \leqslant J \leqslant \text{length(user history)}$. We show an illustration of this state-selection scheme in Fig.2. We show some simulations with the choice of $J$ in Sec.5 to illustrate its effect on the performance of GRRS. Note that we can sum together the user embeddings with the item embeddings as shown in Alg.2 since all the nodes are represented in the same vector space He et al. (2020).

- **The fully adaptive scheme.** We present this algorithm in detail in Alg.3. This scheme starts by *randomly* initializing the weights of both the GNN and RL blocks and updates their weights in every round, as and when a data tuple $\langle s, a, s', r \rangle$ is obtained. This method is more computationally expensive since all the weights (of the GNN block and the DQN block) are learned in each round, along with the graph evolution. The schematic flow of this method can be found in

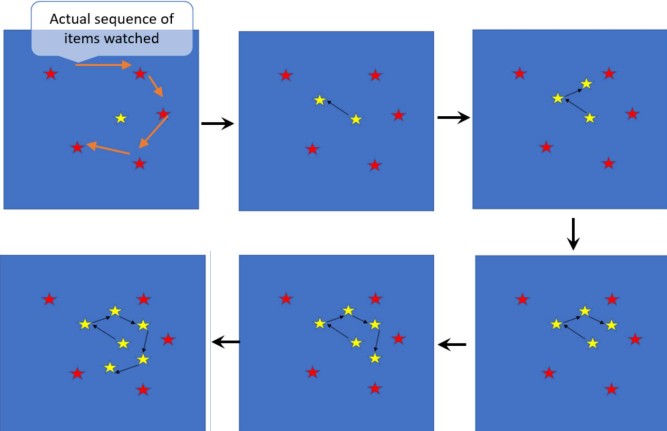

Figure 2: An illustration of the state evolution by the weighted averaging scheme. The red points represent the item embeddings, and the yellow star represents the user embedding. The figures illustrate the effect of the weightings on the user. The user feature moves towards the recently consumed item features.

the appendix in Fig.4. Here, we evolve the graph sequentially as and when an item is recommended to a user. In particular, let at round $t$ of episode $m$, the history available to the agent is $\mathcal{H}_t \equiv \{u^0, i_0^0, r_0^0, u^0, i_1^0, r_1^0, \ldots, u^0, i_{eoe0}^0, r_{eoe0}^0, u^1, i_0^1, r_0^1, \ldots, u^1, i_{eoe1}^1, r_{eoe1}^1, \ldots, u^m, i_0^m, r_0^m, \ldots, u^m, i_{t-1}^m, r_{t-1}^m, u^m\}$, then the graph at round $t$ will be built causally on the history $\mathcal{H}_t$. Then, this graph is passed through the LightGCN layer, where the users and items are associated with a feature vector, which will, over time, tend to encapsulate the properties of the neighbors of the individual users and items. The user $u_t^m$ will be similarly represented by a feature vector (embedding) $s_t \in \mathbb{R}^d$. We use this vector $s_t$ as the state of the system at round $t$, which is then fed to the RL block for item recommendation.

## 4.2 RL BLOCK

We employ a Deep Q-network (DQN)Mnih et al. (2013) for the RL agent. DQN is a Q-learning algorithm where the Q-function is approximated by a Deep Neural Network (DNN). DQN is a *off-policy, bootstrapped* algorithm, which makes it suitable to use when the episodes' initial states are chosen arbitrarily Sutton et al. (2000).

### 4.2.1 "OFFLINE TRAINING" ISSUE IN RL.

An infamous consequence of using reinforcement learning on static logged datasets is the offline training problem (see, e.g., Prudencio et al. (2023) and references therein). The data is usually cast into tuples of the form $\langle s, a, s', r \rangle$, where $s$ is the current state, $a$ is the action taken, $s'$ is the next state observed, and $r$ is the immediate reward obtained. Standard RL algorithms assume that the action taken $a$ is drawn *according* to the current behaviour policy (the policy given by the current configuration of the DQN agent). However, since the data is collected with an arbitrary unknown policy, this results in a *distributional shift* effecting the performance of the system adversely.

In order to deal with this issue, we create an ***artificial** replay buffer* (ARB). We can afford to create such an entity due to the fact that the transition kernel is assumed to be deterministic. This artificial replay buffer is defined as follows. Let the current state be $s_t$. Let $\tilde{a}_t$ be defined as the action (item) recommended by the RL agent. Let us define the reward $\tilde{r}_t$ as $+1$ if the actual action $a_t$ (as read from the dataset) matches $\tilde{a}_t$; else the reward is defined as $-1$ (we assume a negative reward in case of a miss in order to avoid the system getting stuck in a local maxima). Next, let $\tilde{s}_t$ be defined as the next state of the user on application of action $\tilde{a}_t$. The tuple so formed $\langle s_t, \tilde{a}_t, \tilde{s}_t, \tilde{r}_t \rangle$ is then pushed to the ARB. After observing the *actual* item $a_t$ (from the dataset), the actual next state $s'_t$ can again be computed and is used as the initial state for the next round $t+1$, since Q-learning is an *off-policy* algorithm and hence the initial state in each round can be chosen *arbitrarily* (Sutton & Barto, 2018).

---

**Algorithm 1** GNN-RL based Recommender System (GRRS)

---

1: **Input:** Set of episodes $\mathcal{E}$ which contains trajectories of transitions of the form $(s, a, s', r)$ (see Sec. 5.1, pretrained GNN embedding matrix E).
2: **Initialize:** Action-value $Q-$function with random weights $w$.
3: **for** $e \leftarrow 0$ **to** num-epochs **do**
4:     **for all** episode $i$ in $\{0, 1, 2, \ldots, \text{num-episodes}\}$ **do**
5:         Observe user $u^i$.
6:         Select $s_0^i = State - Selection(\text{E}, u^i, user - history)$ (Alg.2)
7:         **for** $t \leftarrow 0$ **to** length-of-episode **do**
8:             Select action $a_t^i$ according to an $\varepsilon - greedy$ rule on the current estimated Q-values.
9:             Simulate the next state $\tilde{s}_{t+1}^i$ according to $State - Selection(s_t^i, user - history \cup \{a_t^i\})$.
10:             **if** actual-item-chosen $== a_t^i$ **then**
11:                 $\tilde{r}_t^i = 1$
12:             **else**
13:                 $\tilde{r}_t^i = -1$
14:             **end if**
15:             Store transition $(s_t^i, a_t^i, \tilde{s}_{t+1}^i, \tilde{r}_t^i)$ in ARB.
16:             Observe the actual-movie-seen $a'^i_t$
17:             Obtain the next state $s'^i_{t+1}$ according to $State - Selection(s_t^i, user - history \cup \{a'^i_t\})$.
18:             Sample random mini-batch of size $B$ transitions $(s_j^i, m_j^i, s_{j+1}^i, r_j^i)$ from ARB, $j \in [B]$.
19:             **if** $s_{j+1}^i$ is a terminal state **then**
20:                 Define $y_j^i := r_j^i$.
21:             **else**
22:                 Define $y_j^i := r_j^i + \gamma \max_{a'} Q(s_{j+1}^i, a')$.
23:             **end if**
24:             Take a gradient descent step on $(y_i - Q(s_i, m_i); w)^2$ with step-size $\alpha$.
25:             $s_{t+1}^i \leftarrow \tilde{s}_{t+1}^i$.
26:             $user - history \leftarrow user - history \cup \{a'^i_t\}$).
27:         **end for**
28:     **end for**
29: **end for**

---

**Algorithm 2** State-Selection

---

1: **Input:** GNN-pre-trained embedding matrix E, user ID, user-history $\mathcal{U}$
2: **if** user-history **is** $\emptyset$ **then**
3:     Assign the node embedding of the user read off from the pre-trained GNN block, i.e., $s = \text{E}(user)$.
4: **else**
5:     $s = \text{E}(user) + \sum_{k=1}^{|\mathcal{U}|} \frac{1}{k} \text{E}(item(k))$
6: **end if**
7: **RETURN** $s$.

---

**Note 2.** *We prefer to use the pre-trained-GNN network as a state extractor for the following reasons:*

1. *Reduced time-complexity. The pretraining of the GNN block can be done offline. Hence, during the actual online process, the training is to be done only over the parameters of the DQN block, which reduces the time complexity enormously.*

2. *Finite state space. Using a pre-trained GNN block and only picking the embeddings as initial states in every episode provides working in only a finite state space. This helps as the standard Q-learning algorithm (i.e., without function approximation) is well-known to converge to the (true) optimal Q-values Watkins & Dayan (1992) when the state space is finite (and with appropriately cooled learned rate). Hence, we expect that the DQN network gets closer to the (true) optimal Q-function, leading to enhanced item-recommendations.*

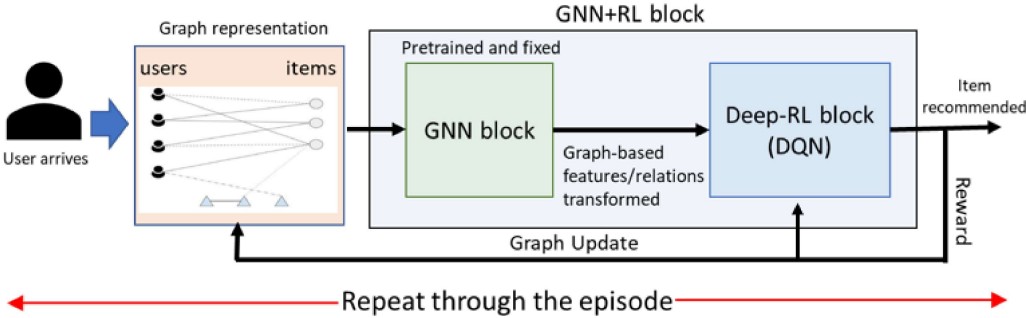

Figure 3: A schematic of GRRS with pretrained GNN block (warm network).

---

**Algorithm 3** Fully adaptive GRRS

---

1: **for all** epoch $e$ **in** `num-epochs` **do**
2:    **Initialize** empty graph $\mathcal{G}_0$, initialize all node embeddings as randomly drawn from normal distribution $\mathcal{N}(0, \sigma^2)$ and call the embedding matrix $\tilde{\mathbb{E}}$.
3:    Observe user ID $u^i$
4:    **for all** episode $i$ **in** $\mathcal{E}$ **do**
5:      Select $s_0^i$ from $\tilde{\mathbb{E}}$
6:      **for** $t \leftarrow 0$ **to** `length-of-episode` **do**
7:        Select action $a_t^i$ according to an $\varepsilon - greedy$ rule on the current estimated Q-values.
8:        $\tilde{\mathcal{G}}^t \leftarrow \mathcal{G}^t \cup \{(s_t^i, a_t^i)\}$.
9:        Simulate the next state $\tilde{s}_{t+1}^i$ by passing $\tilde{\mathcal{G}}^t$ through GCN block.
10:       **if** `actual-item-chosen` $== a_t^i$ **then**
11:         $\tilde{r}_t^i = 1$
12:       **else**
13:         $\tilde{r}_t^i = -1$
14:       **end if**
15:       Store transition $(s_t^i, a_t^i, \tilde{s}_{t+1}^i, \tilde{r}_t^i)$ in ARB.
16:       Observe the actual-movie-seen $a_t'^i$
17:       $\mathcal{G}^t \leftarrow \mathcal{G}^t \cup \{(s_t^i, a_t'^i)\}$.
18:       Simulate the next state $s_{t+1}^i$ by passing $\mathcal{G}^t$ through GCN block.
19:       Sample random mini-batch of size $B$ transitions $(s_j^i, m_j^i, s_{j+1}^i, r_j^i)$ from ARB, $j \in [B]$.
20:       **if** $s_{j+1}^i$ is a terminal state **then**
21:         Define $y_j^i := r_j^i$.
22:       **else**
23:         Define $y_j^i := r_j^i + \gamma \max_{a'} Q(s_{j+1}^i, a')$.
24:       **end if**
25:       Update weights of GNN and hence $\tilde{\mathbb{E}}$; Update weights of DQN.
26:       $s_t^i \leftarrow s_{t+1}^i$.
27:      **end for**
28:    **end for**
29: **end for**

---

## 5 EXPERIMENTAL SETUP

### 5.1 DATASETS AND PREPROCESSING

We work with three datasets. The statistics of these datasets are highlighted in Table 1. Each of the above datasets is collection of user-item interactions. These comprise of four essential quantities that we require:"user ID", "item ID", "rating" and "timestamp". We now describe the method we use to convert these static datasets to ones that can be fed to the RL agent for learning. We convert

Table 1: Statistics of the datasets used.

| Dataset | Number of users | Number of items | Number of interactions |
|---|---|---|---|
| Goodreads (sampled) | 663 | 1000 | 73270 |
| MovieLens-1m (sampled) | 670 | 3883 | 100495 |
| MovieLens-100K | 942 | 1683 | 100029 |

the data into chunks, which we call episodes or sessions interchangeably, characterized by a run of interactions from the same user according to the ascending order of their timestamps. We adhere to the following steps, similar to Dabral et al. (2023):

Step 1. Sort the ratings according to ascending order of timestamp.

Step 2. Starting with the first entry, group the interactions in a single session until there is a change in the user or the time difference between successive timestamps is more than some fixed $l$ units, which we call session-threshold. This marks the end of the current session.

Step 3. GOTO Step 2 and repeat till the end of data.

**Note 3.** *We discard sessions with $< 5$ interactions, as very short sessions may not help in effective learning. Further, we also discard users which have $< 3$ sessions for all our experiments.*

Let us call this set of processed episodes as $\mathcal{E}$.

Creation of train and test set. For every user $u$, we put the first $n_u - 1$ episodes in the train set and the last episode in the test set, where $n_u$ represents the number of sessions of user $u$.

## 5.2 RESULTS

Assume that the agent recommends $K$ items to the user. Recall the definition of *Recall@K* as:

$$Recall@K := \frac{\text{\# relevant items which are recommended}}{\text{\# relevant items}}$$

For maximum user satisfaction, we aim at maximizing Recall@K. As discussed before, we convert all the datasets into sessions and preprocess them (e.g., we remove very short sessions and users with very few ratings) for all our simulations. Details of all the tunable hyper-parameters can be found in the appendix.

### 5.2.1 ON MOVIELENS DATASETS.

We carried out experiments on the movielens-1m and movielens-100K datasets. In Table2a, we compare the performance of GRRS with the current SOTA algorithms. An interesting fact to note here is that CD-HRNN Dabral et al. (2023) which is very close to the performance of GRRS, uses additional context information of the users and the nodes (e.g., BERT embeddings of movie reviews), which our algorithm does not. GRRS, as presented in this paper, initializes its weight randomly and relies only on the past ratings of the users He et al. (2020). This is, in fact, more robust since additional contexts may sometimes be incorrect and misleading He et al. (2020).

We carry some additional simulations on the movielens-1m dataset as shown in Table3 with different values of $J$ as defined in Sec.4. As expected, the algorithm performs the best with the largest value of $J$. Row number 4 in Table3 presents a scenario where, in order to simulate the next state upon a transition, we weight the item embeddings with an indicator function as shown in eq.2.

$$s = \mathrm{E}(user) + \sum_{k=1}^{|\mathcal{U}|} \frac{1}{k} \mathbb{I}\{r(user, item(k)) \geqslant 3\} \mathrm{E}(item(k)) \tag{2}$$

Our simulation shows that considering only the higher-rated items for state evolution does not necessarily help achieve higher recall. We suspect that for the dataset we consider, users decide to consume a particular item only because they like it, irrespective of the rating they provide. This is probably because MovieLens is *not* an interactively collected dataset. We also carry out simulations with the fully adaptive scheme (rows 5 and 6 in Table 3). Surprisingly, we find that the pre-trained GNN scheme performs better than the fully adaptive scheme in our experiments.

Further, we conduct experiments on the movielens-100K dataset for ablation studies (Table2b). We aimed to see the effect of the individual blocks in our method. We conducted the simulation by first removing the RL block and performing simple dot-product-based recommendations ((He et al., 2020)). On the other hand, we conduct simulations when the GNN block is removed and we use simple 1-0 encoded vectors as state to be fed to the RL block. In both these simulations, we find GRRS comfortably outperforming the rest, reinforcing the advantage of the GNN-RL combination.

Table 2: Simulations

(a) Simulations on Movielens-1m

| Method | Recall@20 |
|--------|-----------|
| Cd-HRNN | 49.56% |
| HRNN | 43.03% |
| **GNN-RL** | **51.07%** |

(b) Simulations on Movielens-100K dataset

| Method | Recall@15 | Recall@20 |
|--------|-----------|-----------|
| GNN followed by dot product | 4.41% | 4.96% |
| 1-0 encoded features + RL | 2.13% | 2.9% |
| **GNN triggered RL Agent** | **19.41%** | **21.25%** |

Table 3: Simulations on Movielens-1m dataset different configurations

| Simulation | Recall@20 |
|------------|-----------|
| Full context across episodes (J = length(user history)) | 51.07% |
| 1-step association in rounds (J = 1) | 48.22% |
| States fixed (No state evolution, J=0) | 38.33% |
| Episode-wise GNN update (Update GNN weights once after every episode) | 44.62% |
| Round-wise GNN update (Update GNN weights once after every round) | 43.07% |
| Reward weighting transitions | 42.16% |

### 5.2.2 On Goodreads dataset.

Next, we conduct simulations on Goodreads, a book rating dataset. This is a vast dataset as it contains ratings of books over a 100-year span. In order to perform simulations within reasonable time complexity, we select the top 1000 most-rated books for our simulations. We compared our results with the HGN method proposed in Ma et al. (2019), which is the current state-of-the-art method for this dataset. We observe that our method outperforms the SOTA comfortably.

Table 4: Simulations on Goodreads dataset

| Method | Recall@10 |
|--------|-----------|
| Hierarchical Gating Networks (HGN) Ma et al. (2019) | 12.63% |
| GNN followed by dot product | 7.13% |
| **GNN triggered RL Agent (our method)** | **21.32%** |

## 6 Conclusion

We introduce a novel approach that integrates Graph Neural Networks (GNNs) and Reinforcement Learning (RL) for the first time in session-based recommendation systems. We have substantiated the necessity for this fusion by presenting a scenario that demonstrates how improved state representations can result in superior recommendations. To validate this claim, we conducted simulations on three benchmark datasets, where we showcased the effectiveness of our methodology in comparison to other state-of-the-art algorithms. Additionally, we address the challenge of offline training in RL algorithms, which arises due to the distributional shift when working with static logged databases.

This work paves the way for a plethora of future directions. One immediate avenue could be to modify the current algorithm for average reward setting in RL since many available datasets do not have a natural *terminal* state. Further, in this work, we employed a simple weighted averaging scheme to obtain the next state (in the pre-trained GNN version of GRRS), which can be improved with a more astute time-encapsulating strategy. Lastly, our future efforts will be focused on addressing the unresolved challenge, specifically the cold-start problem for new users and items.

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
