# OpenReview forum: "GNN-based Reinforcement Learning Agent for Session-based Recommendation"
_ICLR.cc/2024/Conference — ICLR 2024 Conference Withdrawn Submission_

### Official Review · Reviewer_htBw · 2023-10-26

**Soundness:** 4 excellent
**Presentation:** 3 good
**Contribution:** 3 good
**Rating:** 5
**Confidence:** 3

**Summary:**

This paper proposes the GRRS framework to solve the session-based recommendation problem, which leverages both the idea of reinforcement learning and graph neural networks.
Here, the reinforcement learning aims to capture the dynamic nature of session-based recommendation and the GNNs aim to capture the user-specific characteristics. Besides, the authors also create an artificial experience replay buffer to tackle the offline training problem. And extensive experiments on three datasets also demonstrate the effectiveness of the proposed GRRS approach.

**Strengths:**

1. This paper first incorporates the GNNs and reinforcement learning for the session-based recommendation, which can be applied in e-commerce platform and other related fields.
2. This paper innovatively creates an artificial experience replay buffer to tackle the offline training problem in previous RL-based recommendation tasks.
3. Three datasets are collected and for evaluation and detailed algorithms are clearly described.

**Weaknesses:**

1. The experiments are not enough to prove the effectiveness of proposed GRRS approach. And only few baselines are compared with the proposed method.
2. Some mathematical expressions can be simplified, and some settings of baselines should be clarified.
3. The methods can be explained more clearly with more formulations.

**Questions:**

1. In table 2a, table 2b and table 3, different baselines are conducted on different datasets. So it is more reasonable to demonstrate the overall superiority of the proposed method on all datasets with same SOTA baselines. Besides, more session-based recommendation baselines should be considered.
2. The architecture of the experiments is not clear. Baselines, ablation study and parameters sensitivity need to be reorganized for better understanding.
3. The motivations are not clear enough, e.g. the toy example of Figure 1.
4. Some descriptions are vogue, especially in experiments. The GNN-RL, GNN triggered RL Agent and GRRS need to be standardized for understanding.

---

### Official Review · Reviewer_ApTX · 2023-11-08

**Soundness:** 4 excellent
**Presentation:** 2 fair
**Contribution:** 2 fair
**Rating:** 3
**Confidence:** 3

**Summary:**

This paper proposes a novel Graph Neural Network (GNN)-based Reinforcement Learning (RL) agent for session-based recommendations, named GRRS. It tackles the challenge of session-based recommendation, where traditional RL methods face difficulties due to their reliance on one-hot encoded feature vectors, and GNNs do not inherently capture the sequential nature of session-based recommendation data.

**Strengths:**

1.	The combination of GNNs with RL for session-based recommendation can addresses existing gaps in the field.
2.	This paper propose the strategy to handle the offline training problem, as it is a common issue affecting the performance of RL-based systems.
3.	Empirical validation of the proposed method, showing superior results over various datasets.

**Weaknesses:**

1. The contribution of the paper is limited, due to the combination of GNN and RL, which solve the state encoding and the dynamic recommandation problem, respectively.
2. The motivation behind the Assumption 1, 2 and Theorem 1 is confusing. How does Theorem 1 motivation the design of the RL algorithm?
3. The replay buffer in the RL block is common in the off-policy RL algorithm, how does this part solve the critical offline training problem? The difference between traditional replay buffer and this ARB are not obvious.
4. The pretrain of the GNN block is not clear enough to me, the detailed training procedure is needed.
5. The three pseudocodes are too redundant for the presentation of the main paper of ICLR.
6. The baselines are not enough, such as some RL-based algorithms are needed, and the visualization of the experiment is lack.

**Questions:**

see the above comments

---

### Official Review · Reviewer_YFCq · 2023-11-10

**Soundness:** 2 fair
**Presentation:** 2 fair
**Contribution:** 1 poor
**Rating:** 1
**Confidence:** 4

**Summary:**

In this paper, the authors discuss a possible design for using GNNs to obtain user states and train a DQN for session-based recommendation systems. They conducted offline experiments with simulated sessions generated from public datasets and compared the proposed method with two RNN-based recommendation systems.

**Strengths:**

1. This paper discusses how to improve the representation learning for session-based recommendation systems, which is an important research problem.
2. Ablation study is included.

**Weaknesses:**

1. The novelty of this work is limited. The concept of incorporating representation learning with GNNs into RL is not new. Additionally, numerous studies [1, 2, 3] have explored the use of GNNs to enhance user/item representation learning in session-based recommendation or sequential recommendation since 2019.
2. The comparison with existing baseline methods is inadequate. The authors should consider including RL-based and GNN-based methods for session-based recommendation or sequential recommendation in their evaluation.
3. The proposed fully adaptive scheme is conflict with the goal of RL due to its requirement for frequent updates to the graph structure and retraining of the LightGCN model each epoch. Additionally, the network warming technique results in a non-end-to-end system, which limit its effectiveness in dynamic recommendation scenarios.

[1] Wu, Shu, et al. "Session-based recommendation with graph neural networks." AAAI. 2019.
[2] Ma, C., Ma, L., Zhang, Y., Sun, J., Liu, X. and Coates. Memory augmented graph neural networks for sequential recommendation. AAAI 2020
[3] Wang, J., Ding, K., Zhu, Z. and Caverlee, J. Session-based recommendation with hypergraph attention networks. SDM. 2021.

**Questions:**

1. In light of prior research that has also incorporated GNN-based representation learning for session-based recommendation and sequential recommendation, could you please elaborate on the unique contributions of your approach?
2. Given the extensive research on RL-based and GNN-based methods for session-based recommendation, could you explain why you chose only two RNN-based methods as baselines for your evaluation?

---

### Official Review · Reviewer_Hpcc · 2023-11-10

**Soundness:** 1 poor
**Presentation:** 1 poor
**Contribution:** 1 poor
**Rating:** 1
**Confidence:** 3

**Summary:**

The paper proposes some heuristic methods utilizing GNN and RL together for session-based recommendation. The GNN might be helpful to generalize on user states, while RL might help with the long-term optimization.

**Strengths:**

- The motivation example in Fig 1 clearly demonstrates the generalization issue of the one-hot embedding on the user state, which makes GNN based approaches sound reasonable.

- There is, at least, some attempt on showing how the generalization over states translate to Q-value estimate, though the theorem does not make much sense to me.

- There are some empirical studies demonstrate the effectiveness of both of the components.

**Weaknesses:**

- The paper is poorly written. I might be wrong, but i do not think one-hot embeddings is still a common approach in industrial systems, this makes the motivation or the problem this paper tried to solve is not even clear to me.

- Beyond this, the method section is kind of messy. The paper mentions RL as a major component, but the MDP is not even clearly defined! How is the state defined? The paragraph on the top of page 3 does not provide any meaningful information about the specific MDP. The theorem about the smoothness of Q function is even confusing, the GNN helps estimation and generalization if i understand it correctly, and the theorem should make statement about the Q-function estimation error (in evaluation case, or the distance to Q^{*} in optimization case), instead of saying the property of Q^*, the property of Q^* is independent of how you estimate and learn the Q function.

- For the offline RL perspective, there are lots of offline RL algorithms you can utilized, BCQ, IQL, i do not understand the reason of sticking with DQN, and even proposing the ARB. The ARB seems even confusing to me, how it is gonna used to adjust distribution shift? The alg box line 15 and 19 even does not match.

I would suggest the paper to take some effort in the writing, at least to communicate the ideas clearly, the current stage of the paper is not ready for publish at ICLR.

**Questions:**

See Weakness.

---

### Official Review · Reviewer_Zy8x · 2023-11-14

**Soundness:** 2 fair
**Presentation:** 2 fair
**Contribution:** 1 poor
**Rating:** 3
**Confidence:** 5

**Summary:**

The paper introduces a q-learning approach using a GNN as a a state/action representation network. The main idea behind the paper is to cast the problem of recommendation as a MDP and try to model it with reinforcement learning. Given that it is tricky to train recommendation models purely with RL a pretrained GNN is used to model the states. The methods is evaluated on the movielens data against few baselines.

**Strengths:**

Interesting problem and a modern approach to recommendation

Well written and easy to read

**Weaknesses:**

Very limited novelty, these types of algorithms have been previously proposed and a quick search will yield a large number of relevant publications.

The experimental section is very limited. practically no baselines and no abliation study.

Movielens is not a sequential dataset so using it for this types of evaluations is not sound.

**Questions:**

Position your work against the state of the art in RL for recommender systems.